# Teaching–Learning Optimization Algorithm Based on the Cadre–Mass Relationship with Tutor Mechanism for Solving Complex Optimization Problems

**DOI:** 10.3390/biomimetics8060462

**Published:** 2023-10-01

**Authors:** Xiao Wu, Shaobo Li, Fengbin Wu, Xinghe Jiang

**Affiliations:** 1School of Mechanical Engineering, Guizhou University, Guiyang 550025, China; xiao_wu1999@163.com; 2State Key Laboratory of Public Big Data, Guizhou University, Guiyang 550025, China; wfaceboss@163.com (F.W.); jiagxh0716@163.com (X.J.)

**Keywords:** metaheuristic, new learner strategy, cadre–masses relationship strategy, tutor mechanism, complex engineering design problems

## Abstract

The teaching–learning-based optimization (TLBO) algorithm, which has gained popularity among scholars for addressing practical issues, suffers from several drawbacks including slow convergence speed, susceptibility to local optima, and suboptimal performance. To overcome these limitations, this paper presents a novel algorithm called the teaching–learning optimization algorithm, based on the cadre–mass relationship with the tutor mechanism (TLOCTO). Building upon the original teaching foundation, this algorithm incorporates the characteristics of class cadre settings and extracurricular learning institutions. It proposes a new learner strategy, cadre–mass relationship strategy, and tutor mechanism. The experimental results on 23 test functions and CEC-2020 benchmark functions demonstrate that the enhanced algorithm exhibits strong competitiveness in terms of convergence speed, solution accuracy, and robustness. Additionally, the superiority of the proposed algorithm over other popular optimizers is confirmed through the Wilcoxon signed rank-sum test. Furthermore, the algorithm’s practical applicability is demonstrated by successfully applying it to three complex engineering design problems.

## 1. Introduction

Optimization algorithms are a class of mathematical techniques employed to seek the optimal solution for a problem, with the primary focus on either maximizing or minimizing an objective function [1]. Traditional optimization methods encounter various challenges as the scale and complexity increase, such as high costs, low efficiency, long execution times, and a tendency to become trapped in local optima [2]. However, metaheuristic optimization algorithms draw inspiration from natural phenomena and the fundamental characteristics of biological systems, endowing them with the capability to solve a wide range of real-world problems [3]. These metaheuristic algorithms possess numerous advantages, including efficient operation, adaptable flexibility, robust stability, exceptional self-organization capabilities, straightforward implementation, potent parallelism, and seamless integration with other algorithms [4]. A myriad of metaheuristic algorithms have been developed to address diverse optimization problems. These algorithms leverage two essential attributes—exploration and exploitation—to effectively navigate the problem spaces and unveil optimal solutions [5].

Nature-inspired algorithms, also known as methods for simulating biological or physical phenomena to tackle optimization problems, play a crucial role in the field. These approaches can be broadly classified into three main types: evolutionary-based, physical-based, and population-based [6]. Evolutionary algorithms, such as genetic algorithms (GA) [7] and differential evolution (DE) [8], draw inspiration from the principles of evolution in biology. GA emulates natural selection, crossover, mutation, and other biological processes to generate novel solutions, retain superior individuals, and progressively explore the optimal solution. Similarly, DE treats each individual as a vector within an n-dimensional space, utilizing operations like mutation, crossover, and selection to iteratively search for the optimal solution. In practical applications, evolutionary algorithms can be used for various optimization problems, such as combinatorial optimization, nonlinear programming, and function optimization.

Physics-based algorithms are a class of optimization algorithms that leverage simulations of physical phenomena to address problems. These algorithms incorporate various physics models, including partial differential equations, kinetic equations, and probability distributions. An illustrative example is simulated annealing (SA) [9], which emulates the annealing process of materials by gradually reducing the temperature and making decisions regarding accepting or rejecting new states based on changes in energy. This approach enables a comprehensive exploration of optimal solutions. Another notable algorithm is particle swarm optimization (PSO) [10], which views the problem as a group of particles, with each particle representing a potential solution. Through iterative updates utilizing velocity and position mechanisms, PSO aims to discover the global optimum. The efficacy of these algorithms has been demonstrated across diverse domains, such as combinatorial optimization, image processing, and machine learning.

Population-based algorithms belong to a category of bionic algorithms that address optimization problems by simulating the cooperative behaviors observed in natural groups. These algorithms facilitate interactions and collaboration among multiple agents. For instance, the green anaconda optimization (GAO) algorithm [11] is derived from the natural mating and hunting behaviors of male anacondas, specifically their ability to locate female anacondas. Similarly, the egret swarm optimization algorithm (ESOA) [12] takes inspiration from the hunting strategies employed by two egret species—the great egret and the snow egret. The ESOA encompasses three crucial components: a sit-and-wait strategy, an aggressive strategy, and discriminant conditions. Notably, this population-based algorithm has extensive applications across various fields, such as engineering design, bioinformatics, finance, etc.

Similar advantages are shared among these algorithms, which possess the capability to address multiple objective functions and nonlinear constraints without the requirement of resolving the derivative or function continuity of the problem, thereby rendering them applicable to a diverse range of optimization problems.

Teaching–learning-based optimization (TLBO) [13] is a technique that draws inspiration from teaching methods employed in the education process, and simulates the influence of teachers on students. This algorithm, despite having fewer parameters, demonstrates excellent performance across various optimization problems. TLBO, along with its enhanced versions, has shown effectiveness in addressing continuous optimization problems [14], combinatorial problems [15], and real-world engineering problems [16]. However, through our rigorous literature survey, we have identified areas where the results presented in earlier studies can be improved in terms of accuracy, robustness, and convergence of the solutions. For example, the LNTLBO algorithm [17], an improved version of TLBO, integrates a logarithmic spiral strategy and a triangular mutation rule to enhance the learning process. By incorporating the logarithmic spiral strategy during the teacher stage, students can actively seek guidance from their teachers, thereby accelerating convergence speed. Moreover, the adoption of a new triangular mutation learning mechanism further improves the learners’ exploration and exploitation abilities. Another approach, the artificial bee colony and teaching–learning-based optimization (ABC-TLBO) algorithm [18], revamps the search strategy of both employed and onlooker bees. It builds upon the basic ABC framework and incorporates TLBO in the observer bee stage to enhance the algorithm’s exploitation capabilities. To improve the quality of the solutions, a chaotic teaching–learning-based optimization (chaotic TLBO) algorithm [19] is proposed, which adopts different chaos mechanisms and introduces a local search method. These additions aim to improve the overall quality of the solution. In conclusion, there is a significant demand for the improvement of the TLBO algorithm, as it holds tremendous potential to enhance performance and provide more satisfactory solutions for complex problems.

In this paper, a new optimization method TLOCTO (teaching–learning optimization algorithm based on the cadre–mass relationship with tutor mechanism) is proposed as an innovation point for a cadre–mass relationship strategy and a tutor mechanism, improving the teacher phase and learner phase based on the TLBO algorithm. Among them, the teacher phase and new learner phase are mainly designed for global exploration. To maximize the global search, the cadre–mass relationship strategy and the tutor mechanism are mainly applied to the algorithm exploration phase to solve the TLBO algorithm’s premature convergence and tendency to fall into local optimum, The two mechanisms are explored and exploited based on the global optimization performed in the teacher phase and new learner phase, which help to achieve a proper balance between exploration and exploitation of solutions, and to ultimately find the solution with the optimal quality.

The remainder of this paper is organized as follows. Section 2 provides a brief overview of related work. The mathematical model of the proposed algorithm is presented in Section 3. Section 4 presents simulations, experiments, and an analysis of the results. Mechanical engineering design problems are described in Section 5. Finally, Section 6 concludes the paper and outlines future research directions.

## 2. Related Work

The basic TLBO algorithm mainly consists of two roles: teachers and students. Teachers are professionals who engage in teaching and imparting knowledge and skills. They work in schools or other educational institutions using various teaching methods to convey knowledge and skills and guide students’ growth and development. Teachers need not only solidly subject knowledge but also good pedagogical skills and communication skills to effectively teach students. Students are individuals who receive education in schools or other educational institutions. They acquire new knowledge and improve their comprehensive abilities by learning from teachers, high-achieving peers, and each other in various ways.

### 2.1. Teacher Phase

In the teacher phase, the teaching process is simulated to find the solution with the best objective function value in the class. Using Equation (1), a new potential solution is generated.
(1)$$T_{j}^{new}=X_{j}^{i}+\mathrm{rand}\times \left(X_{j}^{\mathrm{best}\;}-T_{F}\times X_{j}^{avg}\right)$$
where $T_{j}^{new}$ and $X_{j}^{i}$ represent the positions of the individual after and before learning; $T_{j}^{new}$ denotes the teacher’s position, which corresponds to the best individual in the population; $X_{j}^{avg}$ signifies the average level of search agents in the population; $T_{F}$ is a teaching factor that determines the change in the value of $X_{j}^{avg}$; and rand represents a random number between 0 and 1. The value of $T_{F}$ can be either 1 or 2, randomly determined according to the probability given by Equation (1), as $T_{F}=round(1+{rand}_{j}\left(0,1\right)\left\{2-1\right\}$.

### 2.2. Learner Phase

Besides gaining knowledge from the teacher, learners can also enhance their understanding through interaction. During mutual learning, a learner can acquire knowledge from a randomly chosen peer with a higher grade. The learner strategy can be expressed as follows:(2)$$S_{j}^{i\;}{}_{\;}=\{\begin{array}{l} X_{j}^{i}+\mathrm{rand}\times \left(X_{j}^{\mathrm{rand}\;}-X_{j}^{i}\right),f\left(X_{j}^{i}\right)<f\left(X_{j}^{\mathrm{rand}\;}\right)\\ X_{j}^{i}+\mathrm{rand}\times \left(X_{j}^{i}-X_{j}^{\mathrm{rand}\;}\right),\;\mathrm{otherwise} \end{array}$$
where $S_{j}^{i\;}{}_{\;}$ is the position of the student and $X_{j}^{\mathrm{rand}\;}$is the position of a learner randomly selected from the class.

## 3. The Proposed TLOCTO

In this section, the inspiration for the proposed method is first discussed. Then, the mathematical model is provided.

### 3.1. Inspiration

The teaching and learning algorithm’s development concept stems from the teaching procedure, and this article will introduce an improved teaching and learning algorithm—a teaching–learning optimization algorithm based on the cadre–mass relationship with the tutor mechanism (TLOCTO), mainly inspired by conventional teaching methods such as teacher instruction, learning from excellent students, setting class leaders, and extracurricular tutoring. In the remaining sections of Section 3, the proposed teaching and learning behaviors will be mathematically modeled to develop an optimizer with satisfactory search performance.

### 3.2. New Learner Strategy

During the learner phase, a member learns from random solutions in the class to generate potential new members. Students have various preferences for learning modes, such as formal communication, group discussions, or presentations, and can learn from both teachers and classmates. They also have the flexibility to adjust their learning mode according to their specific situation. Therefore, this paper introduces a new learning mode, which is described in Equation (3), to enhance the diversity of students’ learning methods.
(3)$$S_{j}^{\mathrm{new}\;}=\{\begin{array}{l} X_{j}^{i}+\mathrm{rand}\times \left[\left(1-\frac{t}{T}\right)\times X_{j}^{\mathrm{rand}\;}+\left(\frac{t}{T}\right)\times T_{j}^{\mathrm{new}\;}-X_{j}^{i}\right],f\left(X_{j}^{\mathrm{rand}\;}\right)<f\left(X_{j}^{i}\right),\\ X_{j}^{i}+\;\mathrm{rand}\times \left[\left(1-\frac{t}{T}\right)\times X_{j}^{\mathrm{rand}\;}+\left(\frac{t}{T}\right)\times T_{j}^{\mathrm{new}\;}-X_{j}^{\mathrm{rand}\;}\right],\;\mathrm{otherwise} \end{array}$$
where $S_{j}^{new}$ is the position of the student and $X_{j}^{rand}\;$ is the position of a learner randomly selected from the class. t and T are the current and maximum number of iterations.

In this scenario, the learning mode initially emphasizes random learning to achieve population diversity and global search. As time goes on, students increasingly rely on communication with teachers to accomplish local exploration.

### 3.3. Assistance Phase

The stage in question is bifurcated into two distinct strategies: the cadre–mass relationship strategy and the tutor mechanism, both of which have been primarily devised to facilitate regional exploration. Nevertheless, an overabundance of mechanisms can potentially undermine the efficacy of selective development. Hence, in this scheme, two mechanisms are used to further solve the initial position obtained previously, and the strategy with smaller results is finally selected as the solution for the optimal position.

#### 3.3.1. Cadre–Mass Relationship Strategy

If learners learn from everyone around them, which is an inclusive approach to learning, it will inevitably exert an impact on their learning efficiency, and this impact can be either positive or negative depending on the quality and relevance of the information they receive. Therefore, in the class, teachers will generally set students with good academic performance, strong learning ability, and high learning efficiency as class cadres to play exemplary roles. Class cadres serve as a bridge between students and teachers, and their cooperation with teachers can allow the teaching to receive good results. Student cadres are the core of student groups. They are charismatic, influential, and cohesive, which can unite students to become outstanding. The process can be described by Equation (4).
(4)$$S_{j}^{\mathrm{Cadres}\;}=\varpi \times T_{j}^{\mathrm{new}\;}-\mathrm{rand}\times H_{1}\times S_{j}^{\mathrm{new}\;}-H_{2}\times \mathrm{Levy}(D)+\mathrm{rand}\times H_{1}$$
where $S_{j}^{\mathrm{Cadres}\;}$ is represents the student cadres, the current solution of iteration j. ϖ represents the quality function used to balance the search strategy, which can be calculated by Equation (5). *H*_1_ represents the influencing factors in the search for class cadres, which is defined by Equation (6). *H*_2_ is a decreasing value from ε to 0, indicating that knowledge acquisition increases along with multiple links, such as teaching by teachers, learning led by class leaders, and discussion among students. This efficiency can be defined by Equation (7).
(5)$$\varpi =\frac{H_{1}}{t^{{\left(1-T\right)}^{2}}}$$
(6)$$H_{1}=\mu \times \mathrm{rand}-1$$
(7)$$H_{2}=\varepsilon \times \left(1-\frac{t}{T}\right)$$

Levy(D) is the Levy selection distribution function [20], defined by Equation (8).
(8)$$\mathrm{Levy}(D)=s\times \frac{\tau \times \omega }{|v{|}^{\frac{1}{\varphi }}}$$
where *μ* and *ε* are a number randomly selected between [1, 10], s is 0.01, and τ and ν are randomly selected numbers in the range of [0, 1]. *ω* is defined by Equation (9).
(9)$$\omega =\frac{\Gamma (1+\kappa )\times \sin\left(\frac{\pi \kappa }{2}\right)}{\Gamma \left(\frac{1+\kappa }{2}\right)\times \kappa \times 2^{\left(\frac{\kappa -1}{2}\right)}}$$
where the *κ* value is 1.5.

#### 3.3.2. Tutor Mechanism

The tutor mechanism is a new mechanism for students to utilize to look for teachers in other teaching institutions in order to improve their knowledge. Applying this principle to this algorithm can expand the original search space and discover agents with better performance outside the original population. This will greatly increase the likelihood of an optimal solution, enriching population diversity and enhancing intelligence capabilities. In each generation, let $S_{j}^{i}\in [L_{G}^{l},L_{G}^{u}]$ be a point in a D-dimensional space, where the bound vectors $L_{G}^{l}$ = [$L_{1}^{l}$, $L_{2}^{l}$, …, $L_{D}^{l}$]T and $L_{G}^{u}$ = [$L_{1}^{u}$, $L_{2}^{u}$, $L_{3}^{u}$, …, $L_{D}^{u}$]T are updated as:(10)$$\left\{ \begin{array}{l} L_{j}^{l}=Min\left(\left[S_{1j}^{\mathrm{new}\;},S_{2j}^{\mathrm{new}\;},S_{3j}^{\mathrm{new}\;},\ldots ,S_{nj}^{\mathrm{new}\;}\right]\right)\\ L_{j}^{u}=\mathrm{Max}\left(\left[S_{1j}^{\mathrm{new}\;},S_{2j}^{\mathrm{new}\;},S_{3j}^{\mathrm{new}\;},\ldots ,S_{nj}^{\mathrm{new}\;}\right]\right) \end{array}\right.$$
where j = 1, 2, 3, …, D. Defining $S_{j}^{Tu}={\left[S_{1}^{Tu},S_{2}^{Tu},S_{3}^{Tu},\ldots ,S_{N}^{Tu}\right]}^{T}$ as a tutor mechanism individual at the current generation j, it can be defined by the tutor mechanism as Equation (11).
(11)$$S_{j}^{\mathrm{Tu}}=S_{j}^{new}+y_{n+1}^{\alpha }\times \left(y_{n+1}^{\beta }\times \left(L_{j}^{u}+L_{j}^{1}-S_{j}^{new}\right)-S_{j}^{\mathrm{new}\;}\right)$$
where $y_{n+1}^{\alpha }$ and $y_{n+1}^{\beta }$ are defined by Equation (12).
(12)$$y_{n+1}=\;\mathrm{mod}\;\left(\delta \times y_{n}\left(1-y_{n}\times \cos\left(\arccos\left(y_{n}\right)\right)\right)\times 10^{4},1\right)$$
where $y_{n}$ ∈ [0, 1], δ is the control parameter and δ ∈ [0, +∞). From the above formula, it can be concluded that the result of $y_{n+1}\in (0,1)$.

This article presents the TLOCTO algorithm, outlined in Algorithm 1, and Figure 1 illustrates the flow chart of TLOCTO. The algorithm comprises six steps, which are summarized as follows:

Population and parameters are initialized. The maximum number of iterations (Tmax) is set to 500 and the total particle size (N) to 30, then all agents are randomly initialized. Fitness values are calculated, then evaluated for each agent based on the objective function. The positions are then updated. In the teacher phase, learner phase, cadre–mass mechanism, and tutor mechanism, the solution is continuously optimized by updating the position of each agent. For the boundary check, it must be ensured that each agent’s position remains within the boundaries of the search space. The global best solution, the current best solution, and its fitness value in each iteration are updated. For the termination criterion, the above steps are repeated until the termination condition is met and the output represents the global best solution and its fitness value.
**Algorithm 1:** The framework of the TLOCTO algorithm1: Initialize the solution’s positions of population N randomly;2: Set the maximum number of iterations (Tmax) and other parameters;3: For t = 1 to Tmax do;4: Calculate the average of the population;5: Select the teacher;6: Calculate the fitness function for the given solutions using Equation (1);7: Find the best solution position and fitness value so far;8: For i = 1 to N do;9: Update the individual position using Equation (2);10: Update the individual position using Equation (3);11: Compare and select the one that generates the smaller value as the update position;12: For i = 1 to N do;13: Update the individual position using Equation (4);14: Update the individual position using Equation (11);15: Calculate the fitness values Fitness (SjCadres ) and Fitness (SjTu); 16: If Fitness (SjCadres ) < Fitness (SjTu), then17: Obtain the best position and the best fitness value of the current iteration using Equation (4);18: else;19: Obtain the best position and the best fitness value of the current iteration using Equation (11);20: end if;21: end for;22: end for;23: Return the best solution.

### 3.4. Computational Complexity Analysis

The computational complexity of the TLOCTO algorithm primarily depends on three factors: the initialization process, the evaluation of the fitness function, and the updating of the solutions. The complexity of the initialization process is O(N), where N represents the size of the population. The fitness function depends on the problem, so we will not discuss it here. Finally, the complexity of updating the position is indicated by O(T×N)+O(T×N×D), where T represents the number of iterations and D represents the number of parameters (dimensions) in the problem. Therefore, the computational complexity of the proposed TLOCTO is O(N×(T×D+1)).

## 4. Experimental Results and Detailed Analyses

In this section, we use two types of benchmark functions to investigate the effectiveness of the TLOCTO algorithm. After the qualitative evaluation of the TLOCTO algorithm through standard benchmark functions (the details of these functions can be found in Appendix A) [21], the algorithm was then subjected to testing to assess its efficacy in terms of solving numerical problems. Moreover, the performance of the TLOCTO algorithm in tackling intricate numerical problems was evaluated using the CEC2020 test functions [22]. The TLOCTO was compared to several renowned optimizers, including the artificial bee colony (ABC) [23], genetic algorithm (GA) [7], particle swarm optimization (PSO) [10], grey wolf optimizer (GWO) [24], coati optimization algorithm (COA) [25], and dung beetle optimizer (DBO) [26]. Moreover, TLOCTO was also compared to teaching–learning-based optimization (TLBO) [13]. It is worth noting that these algorithms not only cover recently proposed technologies such as the GWO, DBO, and COA algorithms, but also include classical optimization methods such as the ABC, GA, PSO, and TLBO algorithms. To ensure a fair experimental comparison, the comparison algorithms were executed under identical test conditions. The numerical experiments were conducted using MATLAB 2021b on a computer equipped with an AMD Ryzen 53550H CPU @2.10 GHz and 16 G RAM, running on a 64-bit Windows 10 operating system. Among them, the Wilcoxon rank-sum test [27] was designed with reference to the setting of PlatEMO [28]. The population size was set to N = 30, the maximum number of iterations to T = 500, and 30 independent runs were performed. Additionally, the parameter settings of other counterparts referred to their own settings. It is important to note that the tabular data in this paper are presented in scientific notation.

### 4.1. Qualitative Evaluation

In this section, a qualitative analysis of the TLOCTO algorithm is described, focusing on its convergence behavior, exploration, exploitation, and population diversity. This paper aims to evaluate the performance and characteristics of TLOCTO from a qualitative perspective.

#### 4.1.1. Convergence Behavior Analysis

The benchmark test functions verified TLOCTO’s convergence behavior and analyze the experimental results, as shown in Figure 2. Six functions were chosen for analysis, forming a five-column image. In the first column, the two-dimensional shape of the benchmark function was displayed, helping us to understand the complexity of the problem. The second column showed black points as search agents and a red dot as the global optimum. These agents concentrated near the optimal solution, but were distributed across the search space, demonstrating TLOCTO’s effective exploration ability. The third column presented the average change in fitness values among search agents, starting high and decreasing rapidly, indicating the algorithm’s potential to discover the best value. The fourth column showed the search agent’s trajectory, transitioning from fluctuation to stability. This signified the shift from global exploration to local exploitation and facilitated the process of reaching the global optimal value. Lastly, the fifth column illustrated the convergence curve of the TLOCTO algorithm. In unimodal functions, the curve is smooth and continuously declining, indicating the algorithm’s ability to find the optimal solution. For multimodal functions, the convergence curve descends in steps, indicating the algorithm’s capability to consistently escape local optima and reach the global optimum.

#### 4.1.2. Population Diversity Analysis

Population diversity is significant for the performance of metaheuristic algorithms, and was analyzed by conducting experiments on the suite of classical benchmark functions to compare the population size differences between TLBO and TLOCTO. The computation of population diversity was carried out using a moment of inertia $I_{C}$, demonstrated in Equation (13), while $c_{d}$, represented in Equation (14), indicated the dispersion of the population from its mass center c in each iteration, where the parameter $x_{id}$ denoted the value of the *d*th dimension of the *i*th search agent at iteration [29].
(13)$$I_{C}(t)=\sqrt{\sum_{i=1}^{N}\sum_{d=1}^{D}{\left(x_{id}(t)-c_{d}(t)\right)}^{2}}$$
(14)$$c_{d}(t)=\frac{1}{D}\sum_{i=1}^{N}x_{id}(t)$$

The experimental results are presented in Figure 3, indicating that TLOCTO exhibited a higher level of population diversity compared to TLBO throughout all iterations. This significant discovery suggests that TLOCTO can comprehensively explore the search space and effectively avoid premature convergence and stagnation in local solutions. As a result, it can be inferred that TLOCTO possesses a higher potential to attain the global optimal solution.

#### 4.1.3. Exploration and Exploitation Analysis

By dividing the search process into two stages, namely, exploration and exploitation [30], the metaheuristic algorithm displays its essential characteristic. Balancing these two stages can effectively enhance the algorithm’s efficiency. To achieve this objective, we utilized Equation (15) and Equation (16) to determine the percentages of exploration and exploitation, respectively. Additionally, the dimension-wise diversity measurement was calculated using Equation (17), denoted as Div(*t*). It should be noted that ${Div}_{max}$, representing the maximum diversity in the entire iteration process, was also taken into consideration [29].
(15)$$\mathrm{Exploration}(\%)=\frac{\mathrm{Div}(t)}{\mathrm{Div}}\times 100$$
(16)$$\mathrm{Exploitation}(\%)=\frac{\left|\mathrm{Div}(t)-{\mathrm{Div}}_{\max}\right|}{{\mathrm{Div}}_{\max}}\times 100$$
(17)$$\mathrm{Div}(t)=\frac{1}{D}\sum_{d=1}^{D}\frac{1}{N}\sum_{i=1}^{N}\left|\mathrm{median}\left(x_{d}(t)\right)-x_{id}(t)\right|$$

Dimensional diversity measurement was used in [30] to evaluate the balance of each scheme, and it was concluded that the optimal balance for most functions was over 90% exploitation and less than 10% exploration out of 42 function tests. By observing Figure 4, it becomes apparent that the TLOCTO algorithm displayed exceptional outcomes, surpassing 90% exploitation in all of these assessment functions. Such an observation suggests that the TLOCTO algorithm has effectively attained a desirable equilibrium between the processes of exploration and exploitation within the search domain, thereby resulting in an optimal performance. Specifically, the methodology employed in the TLOCTO algorithm incorporates a dynamic balance between the exploratory and exploitative aspects, which subsequently yields remarkable benefits in terms of circumventing local optima and precluding premature convergence.

### 4.2. Performance Indicators

This paper utilizes two statistical tools, specifically the mean value and the standard deviation [31]. The mathematical formulations for these tools are presented as follows:(18)$$\left\{ \begin{array}{l} \mathrm{AVG}=\frac{1}{P}\sum_{i=1}^{P}f_{i}\\ \mathrm{STD}=\sqrt{\frac{1}{P-1}\sum_{i=1}^{P}{\left(f_{i}-M\right)}^{2}} \end{array}\right.$$
where *P* is the number of optimization experiments, AVG stands for average, and $f_{i}$ represents the optimal value in each independent run.

Furthermore, the Wilcoxon signed rank-sum test [27] was used, with a significance level of α = 0.05, to assess the disparity between TLOCTO and its rivals in this study. Specifically, the outcomes of capturing the minimum fitness function value for each of the 30 independent runs were acquired. Subsequently, the individual probability *p*-value associated with the TLOCTO algorithm and each competitor was separately computed using MATLAB. Ultimately, the decision regarding significant distinctions between algorithms relied on comparing the *p*-value against the significance level α. The symbols applied to the Wilcoxon signed rank-sum test were described as “+”, “−”, and “=”, respectively, to indicate that the comparison algorithms showed significantly superior, inferior, and no significant differences compared with TLOCTO’s algorithm.

### 4.3. TLOCTO’s Performance on the Benchmark Test Functions

This section shows and analyzes the test results of the TLOCTO algorithm and its comparison algorithms on the benchmark test functions.

#### 4.3.1. Comparison Using the Benchmark Test Functions

Functions F1-F7 exhibited a unimodal nature, featuring a solitary global optimum, which facilitated the evaluation of exploitation capability in the meta-heuristic algorithms under examination. The comprehensive analysis presented in Table 1 indicates that, except for F5 and F7, the TLOCTO algorithm consistently surpassed all other compared algorithms when considering the standard values associated with unimodal functions. This consistent superiority establishes the TLOCTO algorithm as the most potent and proficient optimizer among the seven tested unimodal functions, thereby providing compelling evidence of its exceptional exploitation ability.

Multimodal functions, as opposed to unimodal functions, possess multiple local optima that increase exponentially with the problem size, which is determined by the number of design variables. Consequently, these test problems hold great value in assessing the exploration capability of an optimization algorithm. According to the data presented in Table 1, TLOCTO surpassed other optimizers in terms of both average values and standard deviations across 13 out of 16 test functions, specifically multimodal and fixed-dimension multimodal functions F8-F23. Furthermore, the TLOCTO algorithm closely approximated the specified standard values in nearly all functions, with the exception of F12-F14, showcasing its exceptional accuracy and stability. The outstanding performance of TLOCTO on these multimodal functions unequivocally validates its remarkable ability to navigate through and avoid local optima. This ability can be attributed to the utilization of a cadre–mass relationship strategy and tutor mechanism within the algorithm, effectively guiding it towards the global optimum.

Based on the Wilcoxon signed rank-sum test results shown in Table 1 (last line), TLOCTO outperformed GA, GWO, PSO, and TLBO with more than 20 significantly better results (“+”). Additionally, it surpassed COA, DBO, and ABC with 12, 16, and 18 superior results, respectively. In essence, the average goodness percentage of TLOCTO across the 23 benchmark functions was 81.99% (($\sum_{i=1}^{7}{+}_{i})/(23\times 7)\times 100\%$). Overall, the results indicate that the cadre–mass relationship strategy and tutor mechanism strategy effectively enhance TLBO’s optimization capability.

#### 4.3.2. Analysis of Convergence Behavior

TLOCTO’s search agents were observed to extensively explore promising regions of the design space and exploit the most optimal solution. In the initial stages of optimization, the search agents underwent abrupt changes before gradually converging. This ensured the convergence of a population-based algorithm to a point in the search space. Figure 5 presents the convergence curves for TLOCTO and comparative algorithms on some of the 23 standard benchmark functions. These curves reflect the convergence rate, which intuitively measures the improvement in exploration and exploitation. The results imply that TLOCTO competes well with other state-of-the-art meta-heuristic algorithms and exhibits superior convergence accuracy, as is consistent with Table 1.

### 4.4. TLOCTO’s Performance on CEC 2020 Test Functions

The TLOCTO algorithm’s superior performance on simple optimization problems was demonstrated by the mentioned benchmark experiments. Moving on to the next evaluation, we introduce CEC 2020 [22], which presented a challenging test suite aimed at assessing the performance of complex optimization problems. This test suite included a variety of hybrid and composition functions that enabled further evaluation of the TLOCTO algorithm. The benchmark functions, as displayed in Table 2, were categorized into four groups: unimodal function (F1), multimodal shifted and rotated functions (F2-F4), hybrid functions (F5-F7), and composition functions (F8-F10). To assess the TLOCTO algorithm and other comparison algorithms, we utilized the AVG, STD, and Wilcoxon signed rank-sum test according to the experimental setup rules outlined in Section 4. The test results for these algorithms in CEC2020 are presented in Table 3 and Table 4 for problem dimension D, equal to 5 and 10, respectively. For each function, the smallest average value is highlighted in bold font.

#### 4.4.1. Analysis of CEC 2020 Test Function

According to the data presented in Table 3, it can be observed that the TLOCTO algorithm exhibited remarkable performance when dealing with the five-dimensional testing problem. Notably, among the 10 CEC 2020 test functions, the TLOCTO algorithm yielded the minimum fitness value outcomes for seven of them, encompassing single-peaked function (F1), multi-modal shift and rotation functions (F2-F4), and hybrid functions (F5-F7). This implies that TLOCTO possesses a more considerable advantage than other algorithms in terms of resolving non-compound functions. Furthermore, based on the information provided in Table 3 (last row), it can be deduced that TLOCTO surpassed various algorithms, such as ABC, GWO, PSO, GA, COA, DBO, and TLBO in 7, 10, 10, 8, 10, 7, and 7 cases, respectively, out of 10 functions. Additionally, its average excellence rate amounted to 84.29% (($\sum_{i=1}^{7}{+}_{i})/(10\times 7)\times 100\%$). Consequently, the TLOCTO algorithm attained superior outcomes compared to other algorithms.

TLOCTO outperformed other methods when solving 10-dimensional problems, as shown by its high success rates in Table 4. Wilcoxon rank-sum tests revealed TLOCTO’s superiority over rivals like ABC, GWO, PSO, COA, DBO, and TLBO on over 7 functions, with an 88.57% optimization rate on 10 functions. Additionally, Table 4 indicates that TLOCTO achieved the highest ranking in nine test functions, making it a highly promising solution for CEC 2020 10-dimensional problems. Overall, our comprehensive analysis of these results confirmed TLOCTO’s exceptional performance when compared to other algorithms.

#### 4.4.2. Analysis of Convergence Behavior

Figure 6 and Figure 7 display the convergence plots of TLOCTO and other comparison algorithms on the CEC 2020 (5D and 10D) test functions, respectively. The vertical axis of these plots indicates the function’s best fitness value, while the horizontal axis represents the number of function evaluations. Upon analyzing these plots, it becomes evident that TLOCTO demonstrated a faster descent rate and superior optimization ability across all test functions. This can be attributed to TLOCTO’s incorporation of the teaching and learning stage strategy, cadre–mass relationship strategy, and tutor mechanism strategy. These strategies enable TLOCTO to strike a better balance between global exploration and local exploitation capabilities. By utilizing the curriculum teaching strategy in conjunction with the teacher phase and learner phase, the TLOCTO algorithm attains powerful search capabilities. These capabilities are further enhanced by the integration of both the cadre–mass mechanism and tutor mechanism, which facilitate local exploration and ensures the algorithm’s stability, robustness, and high convergence accuracy. The aforementioned mechanisms are evidenced in the test plots through the consistent descent and rapid convergence of the red line segments. The results demonstrate that the proposed TLOCTO algorithm excelled in terms of convergence and global optimization capabilities. Furthermore, the superiority and robustness of TLOCTO are further confirmed.

#### 4.4.3. Analysis of Scalability

Based on the above experimental results, it has been demonstrated that the TLOCTO algorithm exhibits a remarkable performance in terms of competitiveness. In order to further illustrate the superiority of the TLOCTO algorithm, this section compares it with two other types of algorithms on the CEC2020 test function suite (Dim = 20). An autonomous teaching–learning-based optimization algorithm (ATLBO) [32], an improved teaching–learning-based optimization algorithm (ITLBO) [33], a teaching–learning-studying-based optimization algorithm (TLSBO) [34], and an improved TLBO with a logarithmic spiral and triangular mutation (LNTLBO) [17] are all new variants of the TLBO algorithm. Additionally, the self-adaptive spherical search algorithm (SASS) [35] is the champion algorithm in the CEC2020 test function suite competition [36]. Based on the results presented in Table 5, it can be concluded that TLOCTO demonstrated an outstanding performance in terms of its capabilities and achieved most of the optimal results. This highlights the significant research value of TLOCTO. Furthermore, as observed in Figure 8, TLOCTO consistently maintained the best convergence speed and exhibited excellent stability. Therefore, considering the range of tested functions, TLOCTO can be regarded as a reliable choice.

## 5. Mechanical Engineering Application Problems

Most real-world engineering optimization problems are non-linear, with complex constraints [37]. Hence, this section tests the optimization performance of the TLOCTO algorithm, developed for practical applications, by using three renowned mechanical engineering problems. These problems feature multiple equality and inequality constraints, which assess the capability of TLOCTO in terms of optimizing real-world and constrained problems from a constraint handling perspective.

When solving engineering design constraint optimization problems with varying levels of complexity, the death penalty functions [38] can be used to handle solutions not meeting the constraint conditions, and the formula is as follows:(19)$$\begin{array}{l} F(x)=F(x)+z\\ z=\sum_{i=1}^{m}\left(\lambda \cdot {\left(g(i)\right)}^{2}\cdot H(i)\right) \end{array}$$
where z is a penalty term, m presents the number of constraints in the problem, l is the penalty constant, and H(i) is used to identify whether the ith constraint condition is met.

Furthermore, all of the algorithm parameters were set to the same values as in the above experiments, and the population size and maximum iteration numbers for all problems were 30 and 500, respectively. The following sections provide detailed descriptions of the three engineering problems and present all comparative results of these algorithms.

### 5.1. Planetary Gear Train Design Optimization Problem

The main goal of this problem was to minimize the maximum errors in the gear ratio [39] utilized in automobiles. To achieve this, the total number of gear teeth is computed for an automatic planetary transmission system. it is shown in Table A4. In which included total 9-decision variables, first 6-decision variables based on the number of teeth in the gears (N1, N2, N3, N4, N5 and N6), namely 1-6 marked in the figure, which can only take integers values, and rest of 3-discrete variables as modules of the gears (m1 and m2) and number of planet gears (P).

The implementation results of TLOCTO and competitor algorithms in terms of achieving the optimal solution for the planetary gear train design optimization problem are reported in Table 6. In addition, Table 7 provides the corresponding constraint values of these algorithms for this problem.

According to the analysis of reference [40], the planetary gear train design optimization problem is one of the most difficult problems in mechanical engineering. As can be seen from Table 6, in solving such problems, the TLOCTO algorithm not only performed the best out of all comparison algorithms, but also obtained the solution closest to the optimal value of the planetary wheel design problem. Therefore, we can say that the TLOCTO algorithm not only has an excellent ability to solve complex mechanical engineering problems, but also gives full play to its high efficiency and accuracy.

### 5.2. Robot Gripper Problem

For this problem [41], we utilized the difference between the minimum and maximum force generated by the robot gripper as the objective function. This problem comprised seven design variables and six nonlinear design constraints associated with the robot. Mathematically, the robot gripper was a single-degree-of-freedom planning closed-loop mechanism, and the schematic diagram of this problem was simplified to a mechanism composed of three connecting rods and four joints, as shown in Table A5, where $Y_{min}$ represents the minimal gripping object dimension (50 mm), $Y_{G}$ signifies the gripper ends’ maximum displacement range (150 mm), $Y_{max}$ denotes the maximal gripping object dimension (100 mm), $Z_{max}$ indicates the maximum gripper actuator displacement (100 mm), and P represents the gripper’s actuating force (100 N).

The implementation results of TLOCTO and the competitor algorithms in achieving the optimal solution for the robot gripper problem are reported in Table 8. In addition, Table 9 provides the corresponding constraint values of these algorithms for this problem.

As can be seen from Table 8, when solving the robot fixture problem, among them, the best value solved by the TLOCTO algorithm was closest to the best value provided by the literature, which indicates that TLOCTO can solve complex mechanical engineering problems. It is worth noting that the values in the table can reflect that neither the newly proposed COA and DBO algorithms, nor the excellent PSO and ABC algorithms proposed earlier, could solve the robot fixture problem well. This intuitively proves the superiority of the TLOCTO algorithm’s performance and its professional ability to solve the complex problems of robot fixtures.

### 5.3. Speed Reducer Design Problem

In this case, the purpose was to minimize the weight of the speed reducer [42]. Seven variables were considered, including face width (*x*_1_), a module of teeth (*x*_2_), a discrete design variable on behalf of the teeth in the pinion (*x*_3_), the length of the first shaft between bearings (*x*_4_), the length of the second shaft between bearings (*x*_5_), the diameter of the first shaft (*x*_6_), and the diameter of the second shaft (*x*_7_). The resulting optimization problem is shown in Table A6. The implementation results of TLOCTO and its competitor algorithms in terms of achieving the optimal solution for the speed reducer design problem are reported in Table 10. In addition, Table 11 gives the corresponding constraint values of these algorithms for this problem.

As shown in Table 10, the TLOCTO algorithm achieved good results regarding the design of the reducer. It is worth noting that the optimal value of the TLBO algorithm also reached the standard value, but the performance of the average value and standard deviation was still worse than that of the TLOCTO algorithm. The reason for this is that, when compared to the issues established in the preceding two sections, the speed reducer design problem appeared to be comparatively uncomplicated, thereby demonstrating that the TLBO algorithm possesses some outstanding performance attributes. However, this further illustrates that the TLOCTO algorithm exhibits stronger robustness and accuracy.

## 6. Conclusions and Future Works

This study proposes a teaching–learning optimization algorithm based on the cadre–mass relationship strategy with the tutor mechanism (TLOCTO), which is an efficient optimizer for complex optimization problems. It significantly enhances the exploration and repair reply exploitation capabilities of algorithms by combining innovative strategies such as the new learner strategy, the cadre–mass relationship strategy, and the tutor mechanism. Among these, the cadre–mass strategy plays a crucial role in the TLOCTO algorithm by effectively improving the algorithm’s global exploration capability. Additionally, the TLOCTO algorithm introduces the tutor mechanism, effectively addressing the problem of falling into the local optima that plagued the original algorithm. Through the coordination of these mechanisms, the TLOCTO algorithm demonstrated outstanding performance. Moreover, for 53 different test functions, it provided high-quality solutions, showcasing its adaptability and robustness when applied to complex optimization problems. Specifically, a comparative analysis was conducted between the TLOCTO algorithm and seven other optimization algorithms on 23 benchmark test functions and CEC2020 test functions (Dim = 5, 10), demonstrating its remarkable search performance in terms of convergence speed, solution accuracy, and stability. Furthermore, even when compared to the new variant of TLBO and the champion algorithm of the CEC2020 test suite function, TLOCTO still demonstrated strong competitiveness and superior performance on the CEC2020 (D = 20) test function. Furthermore, the TLOCTO algorithm successfully solved three mechanical engineering design problems, confirming its superiority over other optimizers.

The implementation of TLOCTO opens up numerous possibilities for future research. One avenue is to develop a variant of TLOCTO specifically tailored for multi-objective optimization problems and to execute it accordingly. Additionally, we plan to utilize TLOCTO in order to address various practical issues, such as bionic robotics, task assignment for multiple agents, data clustering, and feature selection, among others.

## Figures and Tables

**Figure 1 biomimetics-08-00462-f001:**
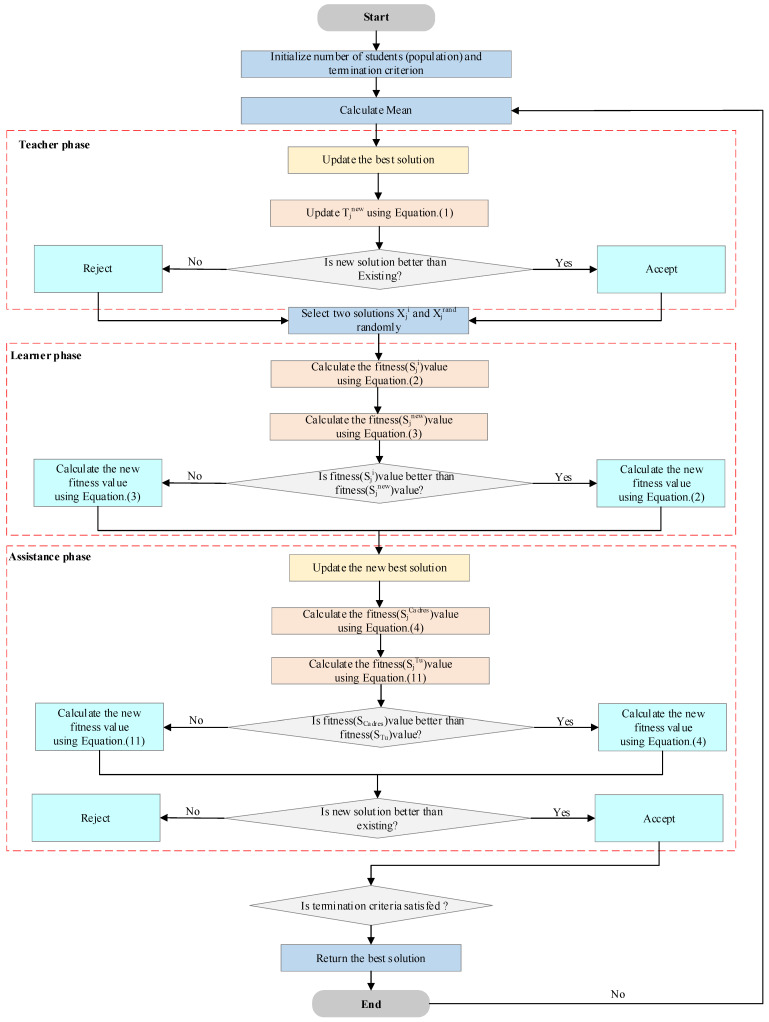
Flowchart of TLOCTO.

**Figure 2 biomimetics-08-00462-f002:**
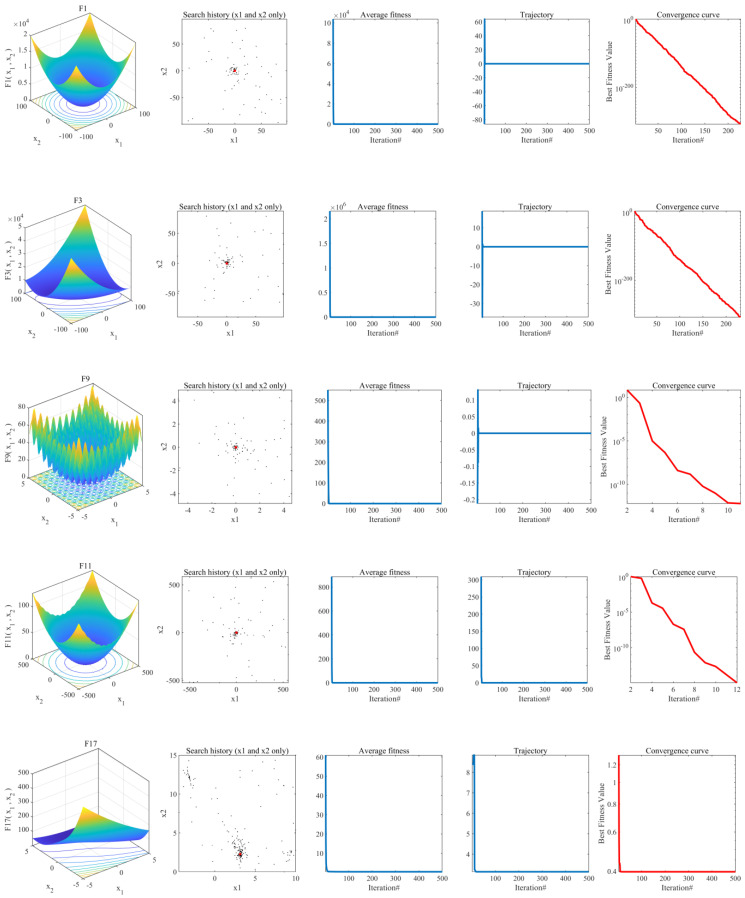
Convergence behaviors of TLOCTO in the search process.

**Figure 3 biomimetics-08-00462-f003:**
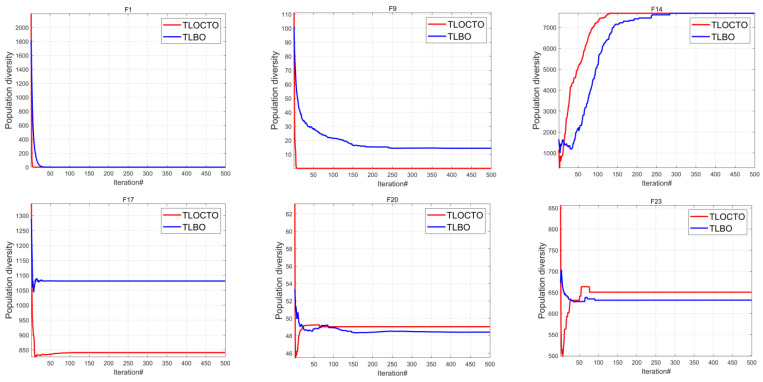
The population diversity of TLBO and TLOCTO.

**Figure 4 biomimetics-08-00462-f004:**
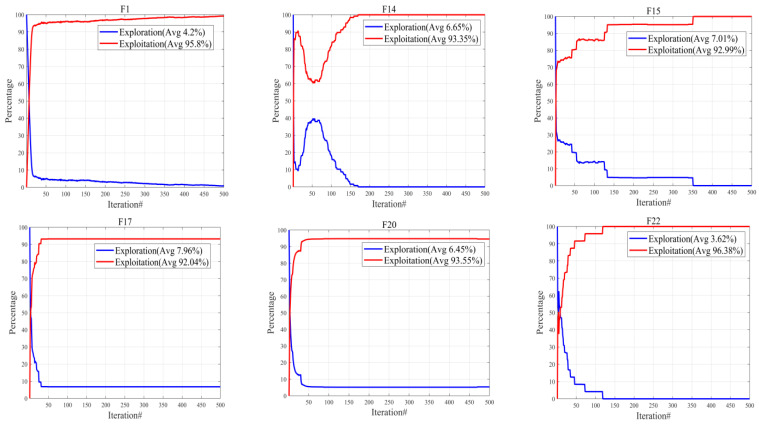
The exploration and exploitation of TLOCTO.

**Figure 5 biomimetics-08-00462-f005:**
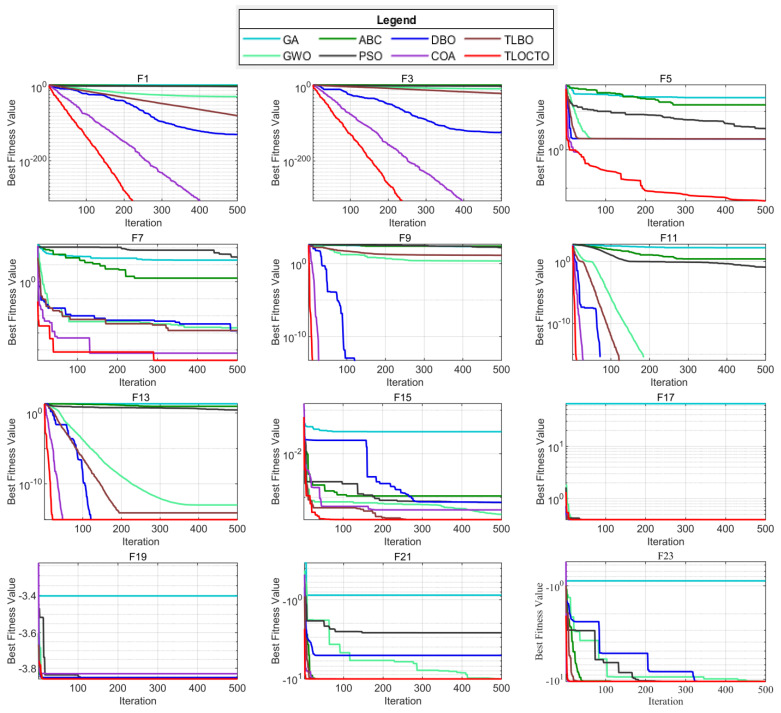
Convergence curves of some benchmark functions.

**Figure 6 biomimetics-08-00462-f006:**
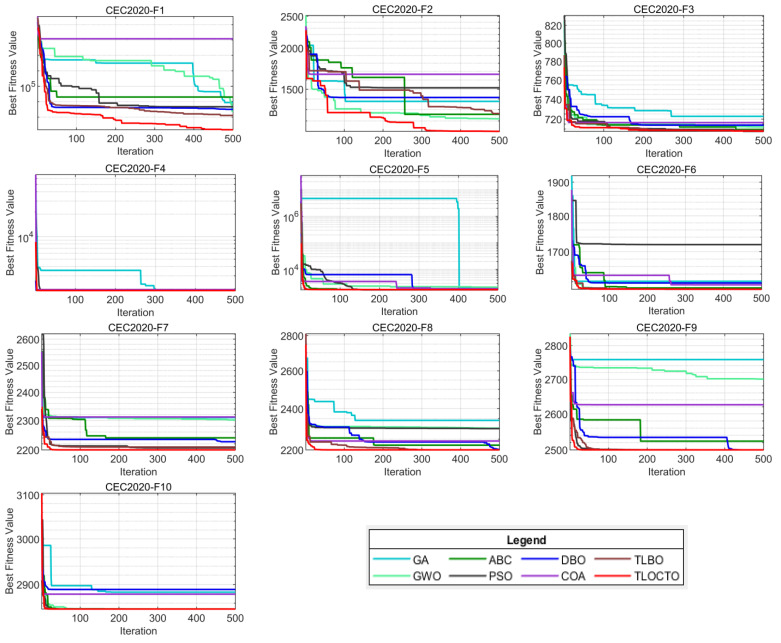
Convergence curves of CEC2020 (5D).

**Figure 7 biomimetics-08-00462-f007:**
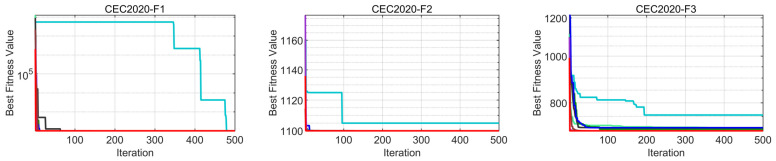
Convergence curves of CEC2020 (10D).

**Figure 8 biomimetics-08-00462-f008:**
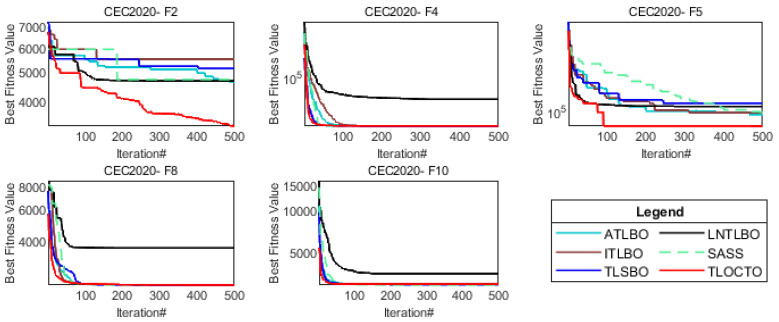
Convergence curves of CEC2020 (20D).

**Table 1 biomimetics-08-00462-t001:** Experimental results of 8 algorithms on the benchmark test functions.

Problem	Metric	TLOCTO	ABC	GWO	PSO	GA	COA	DBO	TLBO
F1	AVG STD	**0.0000 × 10^0^** **0.00 × 10^0^**	8.3930 × 10^0^ 6.86 × 10^0^	8.6133 × 10^−38^ 1.40 × 10^−37^	2.6801 × 10^0^ 1.0384 × 10^0^	1.1609 × 10^0^ 4.16 × 10^−2^	0.0000 × 10^0^ 0.00 × 10^0^	6.0286 × 10^4^ 6.84 × 10^3^	3.3163 × 10^−78^ 1.10 × 10^−77^
F2	AVG STD	**0.0000 × 10^0^** **0.00 × 10^0^**	8.9866 × 10^−1^ 4.11 × 10^−1^	1.5131 × 10^−22^ 1.53 × 10^−22^	4.0965 × 10^0^ 1.0268 × 10^0^	5.8101 × 10^−1^ 1.24 × 10^−2^	1.2046 × 10^−184^ 0.00 × 10^0^	4.3274 × 10^6^ 1.97 × 10^7^	6.1081 × 10^−40^ 5.10 × 10^−40^
F3	AVG STD	**0.0000 × 10^0^** **0.00 × 10^0^**	2.8314 × 10^4^ 8.30 × 10^3^	1.6118 × 10^−1^ 6.14 × 10^−1^	1.8046 × 10^2^ 5.52 × 10^1^	1.6834 × 10^0^ 6.02 × 10^−1^	0.0000 × 10^0^ 0.00 × 10^0^	1.5840 × 10^0^ 6.36 × 10^0^	3.5098 × 10^−17^ 1.06 × 10^−16^
F4	AVG STD	**0.0000 × 10^0^** **0.00 × 10^0^**	8.6368 × 10^1^ 4.76 × 10^0^	2.4439 × 10^−9^ 4.91 × 10^−9^	2.0328 × 10^0^ 2.45 × 10^−1^	2.0040 × 10^−1^ 0.00 × 10^0^	1.2709 × 10^−181^ 0.00 × 10^0^	8.5609 × 10^1^ 4.36 × 10^0^	3.5091 × 10^−33^ 3.01 × 10^−33^
F5	AVG STD	1.8890 × 10^−4^ 4.30 × 10^−4^	4.0777 × 10^3^ 5.00 × 10^3^	2.8415 × 10^1^ 7.62 × 10^−1^	9.3311 × 10^2^ 5.27 × 10^2^	4.2570 × 10^2^ 7.91 × 10^2^	**0.0000 × 10^0^** **0.00 × 10^0^**	2.5716 × 10^1^ 1.78 × 10^−1^	2.6539 × 10^1^ 4.3 × 10^−1^
F6	AVG STD	**0.0000 × 10^0^** **0.00 × 10^0^**	1.4700 × 10^1^ 1.12 × 10^1^	6.6667 × 10^−2^ 2.54 × 10^−1^	2.2603 × 10^0^ 1.03 × 10^0^	3.3333 × 10^−2^ 1.83 × 10^−1^	0.0000 × 10^0^ 0.00 × 10^0^	0.0000 × 10^0^ 0.00 × 10^0^	9.7384 × 10^−3^ 4.1 × 10^−1^
F7	AVG STD	3.6988 × 10^−2^ 3.12 × 10^−2^	2.9306 × 10^−1^ 9.01 × 10^−2^	9.1929 × 10^−2^ 5.26 × 10^−2^	1.7808 × 10^1^ 1.78 × 10^1^	4.8093 × 10^−2^ 1.50 × 10^−2^	**4.8764 × 10^−5^** **3.76 × 10^−5^**	6.0628 × 10^−2^ 5.11 × 10^−2^	1.1936 × 10^−3^ 4.26 × 10^−4^
F8	AVG STD	**−1.1790 × 10^4^** **9.13 × 10^2^**	−9.1626 × 10^3^ 6.99 × 10^2^	−1.5991 × 10^3^ 3.64 × 10^2^	−6.1198 × 10^3^ 1.44 × 10^3^	−1.1152 × 10^4^ 3.29 × 10^2^	−1.2569 × 10^3^ 5.85 × 10^−2^	−8.5886 × 10^3^ 2.12 × 10^3^	−7.4077 × 10^3^ 1.02 × 10^3^
F9	AVG STD	**0.0000 × 10^0^** **0.00 × 10^0^**	1.6257 × 10^2^ 6.13 × 10^1^	0.0000 × 10^0^ 0.00 × 10^0^	1.7389 × 10^2^ 3.71 × 10^1^	2.0327 × 10^0^ 1.21 × 10^0^	0.0000 × 10^0^ 0.00 × 10^0^	2.9850 × 10^−1^ 1.63 × 10^0^	1.6721 × 10^1^ 6.23 × 10^0^
F10	AVG STD	**8.8818 × 10^−16^** **0.00 × 10^0^**	2.6192 × 10^0^ 5.65 × 10^−1^	7.9936 × 10^−15^ 1.32 × 10^−15^	2.6744 × 10^0^ 5.02 × 10^−1^	1.7871 × 10^−1^ 4.16 × 10^−2^	8.8818 × 10^−16^ 0.00 × 10^0^	8.8818 × 10^−16^ 0.00 × 10^0^	6.9278 × 10^−15^ 1.66 × 10^−5^
F11	AVG STD	**0.0000 × 10^0^** **0.00 × 10^0^**	1.0966 × 10^0^ 9.41 × 10^−2^	0.0000 × 10^0^ 0.00 × 10^0^	1.1093 × 10^−1^ 3.73 × 10^−2^	4.5412 × 10^−1^ 1.20 × 10^−1^	0.0000 × 10^0^ 0.00 × 10^0^	0.0000 × 10^0^ 0.00 × 10^0^	3.0152 × 10^−6^ 1.65 × 10^−5^
F12	AVG STD	7.2295 × 10^−7^ 1.13 × 10^−6^	4.0123 × 10^2^ 3.29 × 10^2^	3.1208 × 10^0^ 1.51 × 10^−1^	4.4793 × 10^−2^ 5.51 × 10^−2^	4.1303 × 10^−2^ 3.04 × 10^−2^	**1.5705 × 10^−32^** **5.57 × 10^−48^**	2.4235 × 10^0^ 6.86 × 10^−1^	8.6770 × 10^−5^ 3.01 × 10^−4^
F13	AVG STD	1.8164 × 10^−7^ 2.62 × 10^−7^	1.6327 × 10^3^ 3.62 × 10^3^	2.1180 × 10^0^ 2.40 × 10^−1^	6.3050 × 10^−1^ 2.37 × 10^−1^	2.3183 × 10^−2^ 8.76 × 10^−3^	**1.3498 × 10^−32^** **5.57 × 10^−48^**	6.0204 × 10^−1^ 4.36 × 10^−1^	1.9071 × 10^−1^ 1.48 × 10^−1^
F14	AVG STD	2.0458 × 10^0^ 2.50 × 10^0^	1.6202 × 10^0^ 1.42 × 10^0^	1.2198 × 10^1^ 1.86 × 10^0^	3.0027 × 10^0^ 2.51 × 10^0^	9.9800 × 10^−1^ 2.15 × 10^−11^	9.9800 × 10^−1^ 8.67 × 10^−11^	1.5218 × 10^0^ 1.87 × 10^0^	**9.9800 × 10^−1^** **0.00 × 10^0^**
F15	AVG STD	**3.0979 × 10^−4^** **8.72 × 10^−6^**	1.4522 × 10^−3^ 3.58 × 10^−3^	7.3193 × 10^−3^ 8.39 × 10^−3^	9.2132 × 10^−4^ 2.69 × 10^−4^	4.5409 × 10^−3^ 7.62 × 10^−3^	4.4093 × 10^−4^ 1.20 × 10^−4^	7.7156 × 10^−4^ 2.74 × 10^−4^	3.5310 × 10^−4^ 0.00 × 10^0^
F16	AVG STD	**−1.0316 × 10^0^** **6.65 × 10^−16^**	−1.0316 × 10^0^ 5.53 × 10^−16^	−1.0235 × 10^0^ 7.84 × 10^−3^	−1.0316 × 10^0^ 4.88 × 10^−16^	−1.0316 × 10^0^ 4.94 × 10^−7^	−1.0316 × 10^0^ 1.23 × 10^−4^	−1.0316 × 10^0^ 4.44 × 10^−16^	−1.0316 × 10^0^ 6.71 × 10^−15^
F17	AVG STD	**3.9789 × 10^−1^** **0.00 × 10^0^**	3.9789 × 10^−1^ 0.00 × 10^0^	8.1189 × 10^−1^ 4.91 × 10^−9^	3.9789 × 10^−1^ 0.00 × 10^0^	3.9789 × 10^−1^ 8.38 × 10^−7^	3.9831 × 10^−1^ 8.62 × 10^−4^	3.9789 × 10^−1^ 3.24 × 10^−16^	3.9789 × 10^−1^ 0.00 × 10^0^
F18	AVG STD	**3.0000 × 10^0^** **1.28 × 10^−15^**	3.0000 × 10^0^ 4.24 × 10^−15^	3.2919 × 10^0^ 4.89 × 10^−1^	3.0000 × 10^0^ 6.24 × 10^−14^	3.0000 × 10^0^ 4.84 × 10^−6^	3.0459 × 10^0^ 6.34 × 10^−2^	3.0000 × 10^0^ 1.85 × 10^−14^	3.0000 × 10^0^ 1.39 × 10^−15^
F19	AVG STD	**−3.8628 × 10^0^** **2.71 × 10^−15^**	−3.8628 × 10^0^ 2.46 × 10^−15^	−3.5047 × 10^0^ 3.57 × 10^−1^	−3.8628 × 10^0^ 1.92 × 10^−15^	−3.8628 × 10^0^ 1.74 × 10^−7^	−3.8002 × 10^0^ 7.97 × 10^−2^	−3.8615 × 10^0^ 2.99 × 10^−3^	−3.8628 × 10^0^ 2.71 × 10^−15^
F20	AVG STD	**−3.3146 × 10^0^** **2.79 × 10^−2^**	−3.2744 × 10^0^ 5.92 × 10^−2^	−2.4044 × 10^0^ 2.82 × 10^−1^	−3.2586 × 10^0^ 6.03 × 10^−3^	−3.2744 × 10^0^ 5.92 × 10^−2^	−2.6194 × 10^0^ 3.88 × 10^−1^	−3.2998 × 10^0^ 5.17 × 10^−2^	−3.3100 × 10^0^ 3.62 × 10^−2^
F21	AVG STD	**−1.0153 × 10^1^** **7.01 × 10^−15^**	−6.8147 × 10^0^ 3.68 × 10^0^	−2.4730 × 10^0^ 1.12 × 10^0^	−7.056 × 10^0^ 3.269 × 10^0^	−6.1443 × 10^0^ 3.45 × 10^0^	−1.0153 × 10^1^ 7.07 × 10^−5^	−6.2541 × 10^0^ 2.19 × 10^0^	−1.0153 × 10^1^ 2.92 × 10^−14^
F22	AVG STD	**−1.0403 × 10^1^** **1.32 × 10^−15^**	−7.6146 × 10^0^ 3.54 × 10^0^	−1.3810 × 10^0^ 1.01 × 10^0^	−8.3590 × 10^0^ 3.01 × 10^0^	−7.5984 × 10^0^ 3.31 × 10^0^	−1.0403 × 10^1^ 4.24 × 10^−4^	−7.5244 × 10^0^ 2.77 × 10^0^	−1.0183 × 10^1^ 1.22 × 10^0^
F23	AVG STD	**−1.0536 × 10^1^** **1.89 × 10^−15^**	−8.5397 × 10^0^ 3.38 × 10^0^	−1.3328 × 10^0^ 9.82 × 10^−1^	−9.7550 × 10^0^ 2.17 × 10^0^	−6.0831 × 10^0^ 3.76 × 10^0^	−1.0536 × 10^1^ 8.51 × 10^−5^	−8.0278 × 10^0^ 2.73 × 10^0^	−1.0536 × 10^1^ 3.75 × 10^−3^
(+/−/=)	~	**~**	0/18/5	0/20/3	0/23/0	1/21/1	4/12/7	0/16/7	1/22/0

**Table 2 biomimetics-08-00462-t002:** Descriptions of the benchmark functions from CEC 2020.

Function	Name	Range	*f_min_*
F1 (CEC_01)	Shifted and rotated bent cigar function	[−100, 100]^Dim^	100
F2 (CEC_02)	Shifted and rotated schwefel’s function	[−100, 100]^Dim^	1100
F3 (CEC_03)	Shifted and rotated lunacek bi-rastrigin function	[−100, 100]^Dim^	700
F4 (CEC_04)	Expanded rosenbrock’s plus griewangk’s function	[−100, 100]^Dim^	1900
F5 (CEC_05)	Hybrid function 1 (N = 3)	[−100, 100]^Dim^	1700
F6 (CEC_06)	Hybrid function 1 (N = 4)	[−100, 100]^Dim^	1600
F7 (CEC_07)	Hybrid function 1 (N = 5)	[−100, 100]^Dim^	2100
F8 (CEC_08)	Composition function 1 (N = 3)	[−100, 100]^Dim^	2200
F9 (CEC_09)	Composition function 1 (N = 4)	[−100, 100]^Dim^	2400
F10 (CEC_10)	Composition function 1 (N = 5)	[−100, 100]^Dim^	2500

**Table 3 biomimetics-08-00462-t003:** Comparison results of algorithms on CEC 2020 (5D).

Problem	Metric	TLOCTO	ABC	GWO	PSO	GA	COA	DBO	TLBO
F1	AVG STD	**3.9393 × 10^2^** **3.16 × 10^2^**	3.9926 × 10^3^ 4.21 × 10^3^	1.0928 × 10^8^ 8.11 × 10^7^	1.1321 × 10^8^ 1.88 × 10^8^	4.2118 × 10^3^ 4.17 × 10^3^	9.5389 × 10^8^ 6.45 × 10^8^	3.0308 × 10^3^ 3.67 × 10^3^	3.0856 × 10^5^ 2.95 × 10^5^
F2	AVG STD	**1.1942 × 10^3^** **6.82 × 10^1^**	1.2459 × 10^3^ 1.37 × 10^2^	1.8413 × 10^3^ 2.24 × 10^2^	1.5817 × 10^3^ 1.66 × 10^2^	1.2611 × 10^3^ 1.37 × 10^2^	2.1665 × 10^3^ 2.07 × 10^2^	1.3873 × 10^3^ 1.49 × 10^2^	1.3663 × 10^3^ 1.11 × 10^2^
F3	AVG STD	**7.0722 × 10^2^** **1.44 × 10^0^**	7.0849 × 10^2^ 3.20 × 10^0^	7.3054 × 10^2^ 6.96 × 10^0^	7.1652 × 10^2^ 6.43 × 10^0^	7.0796 × 10^2^ 2.22 × 10^0^	7.5890 × 10^2^ 7.34 × 10^0^	7.1223 × 10^2^ 3.56 × 10^0^	7.1670 × 10^2^ 4.33 × 10^0^
F4	AVG STD	**1.9002 × 10^3^** **1.02 × 10^−1^**	1.9006 × 10^3^ 3.04 × 10^−1^	1.9083 × 10^3^ 3.92 × 10^0^	3.1673 × 10^3^ 5.93 × 10^3^	1.9003 × 10^3^ 1.92 × 10^−1^	1.3707 × 10^4^ 1.38 × 10^4^	1.9013 × 10^3^ 1.15 × 10^0^	1.9008 × 10^3^ 3.40 × 10^−1^
F5	AVG STD	**1.7051 × 10^3^** **7.07 × 10^0^**	1.7364 × 10^3^ 4.72 × 10^1^	6.5142 × 10^5^ 6.29 × 10^5^	7.8483 × 10^3^ 6.66 × 10^3^	1.7394 × 10^3^ 5.00 × 10^1^	3.3860 × 10^6^ 3.96 × 10^6^	2.0135 × 10^3^ 7.92 × 10^2^	1.8101 × 10^3^ 4.17 × 10^1^
F6	AVG STD	**1.6008 × 10^3^** **4.29 × 10^−1^**	1.6025 × 10^3^ 7.38 × 10^0^	1.6927 × 10^3^ 8.03 × 10^1^	1.6455 × 10^3^ 5.27 × 10^1^	1.6115 × 10^3^ 3.07 × 10^1^	1.8093 × 10^3^ 1.04 × 10^2^	1.6055 × 10^3^ 1.19 × 10^1^	1.6067 × 10^3^ 6.44 × 10^0^
F7	AVG STD	**2.1002 × 10^3^** **3.13 × 10^−1^**	2.1004 × 10^3^ 3.01 × 10^−1^	2.1816 × 10^3^ 7.25 × 10^1^	2.1225 × 10^3^ 2.68 × 10^1^	2.1034 × 10^3^ 1.02 × 10^1^	2.2411 × 10^3^ 8.05 × 10^1^	2.1017 × 10^3^ 6.02 × 10^0^	2.1008 × 10^3^ 1.96 × 10^−1^
F8	AVG STD	2.2461 × 10^3^ 5.25 × 10^1^	2.2390 × 10^3^ 4.70 × 10^1^	2.3204 × 10^3^ 3.27 × 10^1^	2.2856 × 10^3^ 6.46 × 10^1^	2.2748 × 10^3^ 5.00 × 10^1^	2.4921 × 10^3^ 1.53 × 10^2^	2.2433 × 10^3^ 4.45 × 10^1^	**2.2358 × 10^3^** **2.72 × 10^1^**
F9	AVG STD	2.5181 × 10^3^ 5.58 × 10^1^	2.5520 × 10^3^ 8.72 × 10^1^	2.7130 × 10^3^ 8.62 × 10^1^	2.6786 × 10^3^ 9.23 × 10^1^	2.5883 × 10^3^ 1.13 × 10^2^	2.7258 × 10^3^ 6.49 × 10^1^	**2.5000 × 10^3^** **2.65 × 10^−4^**	2.5357 × 10^3^ 6.33 × 10^0^
F10	AVG STD	2.8391 × 10^3^ 3.78 × 10^1^	2.8458 × 10^3^ 8.65 × 10^0^	2.8575 × 10^3^ 7.03 × 10^0^	2.8736 × 10^3^ 2.68 × 10^1^	2.8394 × 10^3^ 2.14 × 10^1^	2.9488 × 10^3^ 5.51 × 10^1^	2.8500 × 10^3^ 2.01 × 10^1^	**2.8247 × 10^3^** **1.18 × 10^1^**
(+/−/=)	~	**~**	0/7/3	0/10/0	0/10/0	0/8/2	0/10/0	1/7/2	2/7/1

**Table 4 biomimetics-08-00462-t004:** Comparison results of algorithms on CEC 2020 (10D).

**Problem**	**Metric**	**TLOCTO**	**ABC**	**GWO**	**PSO**	**GA**	**COA**	**DBO**	**TLBO**
F1	AVG STD	**2.6402 × 10^3^** **2.96 × 10^3^**	3.9681 × 10^3^ 3.59 × 10^3^	2.0448 × 10^9^ 6.82 × 10^8^	6.0622 × 10^9^ 3.51 × 10^9^	1.7744 × 10^4^ 1.73 × 10^4^	1.5653 × 10^10^ 6.10 × 10^9^	1.2644 × 10^6^ 4.50 × 10^6^	1.9777 × 10^8^ 9.19 × 10^7^
F2	AVG STD	**1.5719 × 10^3^** **2.81 × 10^2^**	2.4585 × 10^3^ 6.75 × 10^2^	3.0011 × 10^3^ 3.01 × 10^2^	2.2721 × 10^3^ 3.37 × 10^2^	1.5736 × 10^3^ 2.25 × 10^2^	3.6023 × 10^3^ 3.34 × 10^2^	2.0814 × 10^3^ 3.09 × 10^2^	2.4658 × 10^3^ 2.50 × 10^2^
F3	AVG STD	7.2975 × 10^2^ 7.97 × 10^0^	7.4313 × 10^2^ 2.08 × 10^1^	8.0923 × 10^2^ 1.42 × 10^1^	7.8781 × 10^2^ 3.24 × 10^1^	**7.2642 × 10^2^** **7.30 × 10^0^**	9.0337 × 10^2^ 2.78 × 10^1^	7.5004 × 10^2^ 1.86 × 10^1^	8.0925 × 10^2^ 2.81 × 10^1^
F4	AVG STD	**1.9018 × 10^3^** **8.77 × 10^−1^**	1.9031 × 10^3^ 1.67 × 10^0^	2.2790 × 10^3^ 4.16 × 10^2^	7.2218 × 10^4^ 8.19 × 10^4^	1.9023 × 10^3^ 1.03 × 10^0^	5.2207 × 10^5^ 4.34 × 10^5^	1.9053 × 10^3^ 2.68 × 10^0^	1.9102 × 10^3^ 8.24 × 10^0^
F5	AVG STD	**2.7587 × 10^3^** **9.63 × 10^2^**	3.0432 × 10^5^ 4.52 × 10^5^	6.2118 × 10^5^ 1.33 × 10^5^	7.9091 × 10^5^ 8.38 × 10^5^	4.5979 × 10^5^ 5.39 × 10^5^	7.3010 × 10^6^ 7.23 × 10^6^	1.9503 × 10^4^ 2.02 × 10^4^	1.1536 × 10^4^ 5.47 × 10^3^
F6	AVG STD	**1.6706 × 10^3^** **7.49 × 10^1^**	1.7373 × 10^3^ 1.10 × 10^2^	2.0147 × 10^3^ 9.65 × 10^1^	2.1041 × 10^3^ 1.70 × 10^2^	1.7671 × 10^3^ 1.27 × 10^2^	2.8130 × 10^3^ 2.95 × 10^2^	1.8091 × 10^3^ 1.29 × 10^2^	1.7854 × 10^3^ 7.64 × 10^1^
F7	AVG STD	**2.4853 × 10^3^** **1.66 × 10^2^**	1.1205 × 10^4^ 9.80 × 10^3^	3.0861 × 10^6^ 4.62 × 10^6^	6.6925 × 10^5^ 1.30 × 10^6^	1.4839 × 10^5^ 3.55 × 10^5^	4.0225 × 10^6^ 5.47 × 10^6^	7.8715 × 10^3^ 9.06 × 10^3^	4.9133 × 10^3^ 1.50 × 10^3^
F8	AVG STD	**2.3051 × 10^3^** **1.88 × 10^1^**	2.3073 × 10^3^ 1.48 × 10^1^	2.4639 × 10^3^ 5.96 × 10^1^	2.7308 × 10^3^ 3.94 × 10^2^	2.3100 × 10^3^ 8.25 × 10^−3^	3.6911 × 10^3^ 5.98 × 10^2^	2.3115 × 10^3^ 2.22 × 10^0^	2.4274 × 10^3^ 1.59 × 10^2^
F9	AVG STD	**2.7129 × 10^3^** **8.25 × 10^1^**	2.7553 × 10^3^ 1.70 × 10^1^	2.8194 × 10^3^ 1.20 × 10^1^	2.8307 × 10^3^ 1.02 × 10^2^	2.7581 × 10^3^ 1.50 × 10^1^	2.9661 × 10^3^ 1.08 × 10^2^	2.7751 × 10^3^ 3.88 × 10^1^	2.7689 × 10^3^ 4.82 × 10^1^
F10	AVG STD	**2.9279 × 10^3^** **2.32 × 10^1^**	2.9359 × 10^3^ 2.07 × 10^1^	3.0253 × 10^3^ 4.66 × 10^1^	3.1485 × 10^3^ 1.31 × 10^2^	2.9399 × 10^3^ 2.73 × 10^1^	3.9653 × 10^3^ 3.73 × 10^2^	2.9379 × 10^3^ 6.66 × 10^1^	2.9490 × 10^3^ 1.21 × 10^1^
(+/−/=)	~	**~**	0/7/3	0/10/0	0/9/1	1/6/3	0/10/0	0/10/0	0/10/0

**Table 5 biomimetics-08-00462-t005:** Comparison results of algorithms on CEC 2020 (20D).

Problem	Metric	TLOCTO	ATLBO	ITLBO	TLSBO	LNTLBO	SASS
F1	Ave	1.5872 × 10^5^	1.5285 × 10^6^	4.2403 × 10^3^	1.7599 × 10^3^	8.3267 × 10^9^	**1.6100 × 10^3^**
	Std	3.79 × 10^5^	3.25 × 10^6^	3.95 × 10^3^	2.24 × 10^3^	2.71 × 10^9^	**2.26 × 10^3^**
F2	Ave	**3.2654 × 10^3^**	4.7114 × 10^3^	5.1045 × 10^3^	5.3031 × 10^3^	4.4598 × 10^3^	5.4914 × 10^3^
	Std	**1.52 × 10^2^**	6.72 × 10^2^	4.15 × 10^2^	2.20 × 10^2^	5.89 × 10^2^	2.63 × 10^2^
F3	Ave	8.6336 × 10^2^	8.5826 × 10^2^	8.4488 × 10^2^	**8.2675 × 10^2^**	1.0341 × 10^3^	8.2943 × 10^2^
	Std	4.33 × 10^1^	3.80 × 10^1^	2.88 × 10^1^	**3.89 × 10^1^**	6.54 × 10^1^	8.71 × 10^0^
F4	Ave	1.9353 × 10^3^	1.9204 × 10^3^	1.9163 × 10^3^	1.9117 × 10^3^	1.7668 × 10^4^	**1.9088 × 10^3^**
	Std	1.85 × 10^1^	9.98 × 10^0^	6.88 × 10^0^	3.14 × 10^0^	1.45 × 10^4^	**1.40 × 10^0^**
F5	Ave	**4.4176 × 10^4^**	7.2356 × 10^4^	1.3859 × 10^5^	1.7059 × 10^5^	1.6930 × 10^5^	4.7068 × 10^4^
	Std	**3.71 × 10^4^**	5.66 × 10^4^	1.01 × 10^5^	1.29 × 10^5^	2.16 × 10^5^	4.50 × 10^4^
F6	Ave	**1.6291 × 10^3^**	1.7099 × 10^3^	1.8043 × 10^3^	1.9356 × 10^3^	1.7243 × 10^3^	2.2605 × 10^3^
	Std	**2.31 × 10^−13^**	1.16 × 10^−12^	1.16 × 10^−12^	1.16 × 10^−12^	1.16 × 10^−12^	1.85 × 10^−12^
F7	Ave	**1.6466 × 10^4^**	2.0766 × 10^4^	5.2183 × 10^4^	2.7900 × 10^4^	8.8333 × 10^4^	2.1520 × 10^4^
	Std	**9.40 × 10^3^**	1.51 × 10^4^	4.76 × 10^4^	1.71 × 10^4^	1.47 × 10^5^	1.30 × 10^4^
F8	Ave	**2.3061 × 10^3^**	2.3098 × 10^3^	2.4348 × 10^3^	2.3072 × 10^3^	3.8601 × 10^3^	2.3069 × 10^3^
	Std	**3.74 × 10^0^**	9.77 × 10^0^	7.25 × 10^2^	1.30 × 10^0^	8.36 × 10^2^	9.61 × 10^1^
F9	Ave	2.8775 × 10^3^	2.8706 × 10^3^	2.8565 × 10^3^	**2.8418 × 10^3^**	3.0211 × 10^3^	2.8633 × 10^3^
	Std	3.13 × 10^1^	2.41 × 10^1^	2.78 × 10^1^	**1.52 × 10^1^**	5.75 × 10^1^	4.19 × 10^1^
F10	Ave	**2.9981 × 10^3^**	3.0151 × 10^3^	2.9992 × 10^3^	3.0768 × 10^3^	3.4933 × 10^3^	3.0328 × 10^3^
	Std	**3.10 × 10^1^**	4.33 × 10^1^	3.31 × 10^1^	3.28 × 10^1^	2.77 × 10^2^	3.87 × 10^1^
(+/−/=)	~	**~**	0/6/4	1/6/3	1/5/4	0/7/3	1/5/4

**Table 6 biomimetics-08-00462-t006:** Comparison results for the planetary gear train design optimization problem.

Algorithm	Best	Mean	Worst	Std	$N_{1}$	$N_{2}$	$N_{3}$	$N_{4}$	$N_{5}$	$N_{6}$	p	$m_{1}$	$m_{2}$
TLOCTO	**0.52325**	**0.53592**	**0.53706**	**0.00364**	**32**	**18**	**15**	**19**	**15**	**69**	**4**	**2**	**2**
TLBO	0.53706	0.53877	0.55667	0.00494	22	14	15	17	15	62	3	2	2
DBO	0.54846	1.8 × 10^20^	0.77667	3.66 × 10^20^	26	14	14	19	14	69	3	1.75	1.75
COA	0.55706	7.00 × 10^19^	0.86074	1.44 × 10^20^	17	14	20	17	14	62	3	1.75	1.75
GWO	0.52967	0.55229	0.71000	0.03423	47	24	15	21	14	76	3	2	2
PSO	0.52624	0.54632	0.80573	0.04942	26	17	22	24	14	87	3	2.75	1.75
ABC	0.59312	0.93813	1.76841	0.28424	56	26	19	28	28	103	3	2	1.75

**Table 7 biomimetics-08-00462-t007:** The constraint values of the planetary gear train design optimization problem.

Algorithm	Constraints									
	$g_{1}$	$g_{2}$	$g_{3}$	$g_{4}$	$g_{5}$	$g_{6}$	$g_{7}$	$g_{8}$	$g_{9}$	$g_{10}$
TLOCTO	**−77**	**−80**	**−118**	**−4**	**−14.8553**	**−20.6838**	**−6.54163**	**−1165.89**	**−15**	**−15**
TLBO	−91	−116	−122	−18	−14.6769	−23.2032	−10.2128	−1698.90	−91	−116
DBO	−77	−108	−122	−26	−18.141	−31.1314	−12.0788	−3597.35	−17	−17
COA	−91	−126	−126	−18	−10.3468	−13.8731	−10.3468	−377.447	−91	−126
GWO	−63	−26	−118	−16	−34.9878	−35.3275	−13.8109	−4988.21	−63	−26
PSO	−41	−34	−112	−4	−17.7391	−31.7917	−16.4090	−4383.19	−41	−34
ABC	−9	0	−48	0	−42.5141	−51.2461	−17.9974	−7422.03	−9	0

**Table 8 biomimetics-08-00462-t008:** Comparison results for the robot gripper problem.

Algorithm	Best	Mean	Worst	Std	a	b	c	e	f	l	δ
TLOCTO	**2.77479**	**3.09241**	**3.31269**	**0.15419**	**149.2512**	**132.6060**	**200.0000**	**16.4597**	**149.9069**	**104.6298**	**2.4493**
TLBO	3.01791	3.59001	5.98043	0.59922	143.5842	133.7948	200.0000	9.58430	149.9702	106.9715	2.4607
DBO	3.31432	5.82827	9.34913	1.43645	150.0000	149.4977	198.8000	0.0306	17.01180	124.6755	1.7178
COA	3.79460	2.94 × 10^22^	3.56 × 10^23^	7.68 × 10^22^	147.8050	139.9458	153.5251	7.4652	147.8050	115.7780	2.6540
GWO	3.32379	3.77367	4.54004	0.30097	150.0000	140.7627	176.0328	8.7721	149.1059	118.9015	2.5634
PSO	3.44092	4.16993	9.54223	1.07760	150.0000	111.7020	199.5793	37.1027	144.0012	129.4336	2.7289
ABC	4.39851	8.21286	13.19273	2.44974	147.5731	138.0619	197.6639	6.5379	148.0575	160.0094	2.6131

**Table 9 biomimetics-08-00462-t009:** The constraint values of the robot gripper problem.

Algorithm	Constraints						
	$g_{1}$	$g_{2}$	$g_{3}$	$g_{4}$	$g_{5}$	$g_{6}$	$g_{7}$
TLOCTO	**−32.4842**	**−17.5158**	**−45.1971**	**−4.8029**	**−68225.1644**	**−70.6672**	**−4.6299**
TLBO	−43.0799	−6.9201	−31.3405	−18.6595	−65404.3490	−103.5279	−6.9716
DBO	−49.1101	−0.8899	−43.5540	−6.4460	−74154.8911	−750.1439	−24.6756
COA	−6.3163	−43.6837	−20.6581	−29.3419	−69340.2484	−359.3810	−15.7781
GWO	−21.0329	−28.9671	−26.2605	−23.7395	−70328.4313	−488.4525	−18.9016
PSO	−40.5933	−9.4067	−34.8365	−15.1635	−50358.2696	−1134.8062	−29.4337
ABC	−45.0324	−4.9676	−34.9849	−15.0151	−55941.6010	−4430.9795	−60.0095

**Table 10 biomimetics-08-00462-t010:** Comparison results for the speed reducer design problem.

Algorithm	Best	Mean	Worst	Std	$x_{1}$	$x_{2}$	$x_{3}$	$x_{4}$	$x_{5}$	$x_{6}$	$x_{7}$
TLOCTO	**2994.424**	**2994.424**	**2994.424**	**1.85 × 10^−12^**	**3.50000**	**0.70000**	**17.0000**	**7.30000**	**7.71532**	**3.35054**	**5.28665**
TLBO	2994.424	2994.492	2994.870	0.096	3.50000	0.700002	17.0000	7.30006	7.71532	3.35054	5.28666
DBO	3032.779	3406.531	5735.099	782.939	3.50264	0.70000	17.0000	7.30000	7.77305	3.35332	5.28696
COA	3060.413	4.57 × 10^17^	1.31 × 10^19^	2.38 × 10^18^	3.50022	0.70000	17.0000	7.30000	7.89676	3.35155	5.28631
GWO	3003.825	3011.045	3018.471	3.854	3.50122	0.70002	17.0001	7.77206	7.82642	3.35226	5.28846
PSO	3007.437	3160.023	3363.736	120.986	3.50000	0.70001	17.0000	7.30000	8.30215	3.35054	5.28686
ABC	2549.639	2597.282	2635.205	20.995	5.99485	0.70402	14.4866	7.30748	7.90121	3.49492	5.29177

**Table 11 biomimetics-08-00462-t011:** The constraint values of the speed reducer design problem.

Algorithm	Constraints										
	$g_{1}$	$g_{2}$	$g_{3}$	$g_{4}$	$g_{5}$	$g_{6}$	$g_{7}$	$g_{8}$	$g_{9}$	$g_{10}$	$g_{11}$
TLOCTO	**−2.16 × 10^0^**	**−9.81 × 10^1^**	**−1.93 × 10^0^**	**−1.83 × 10^1^**	**−9.35 × 10^−4^**	**−2.15 × 10^−3^**	**−2.81 × 10^1^**	**0.00 × 10^0^**	**−7.00 × 10^0^**	**−3.74 × 10^−1^**	**−5.00 × 10^−6^**
TLBO	−2.16 × 10^0^	−9.81 × 10^1^	−1.93 × 10^0^	−1.83 × 10^1^	−1.01 × 10^−3^	−2.67 × 10^−3^	−2.81 × 10^1^	−1.43 × 10^−5^	−7.00 × 10^0^	−3.74 × 10^−1^	−6.00 × 10^−6^
DBO	−2.18 × 10^0^	−9.85 × 10^1^	−1.94 × 10^0^	−1.79 × 10^1^	−2.73 × 10^0^	−1.38 × 10^−1^	−2.81 × 10^1^	−3.77 × 10^−3^	−7.00 × 10^0^	−3.70 × 10^−1^	−5.74 × 10^−2^
COA	−2.16 × 10^0^	−9.82 × 10^1^	−1.93 × 10^0^	−1.69 × 10^1^	−9.93 × 10^−1^	−1.96 × 10^−1^	−2.81 × 10^1^	−3.14 × 10^−4^	−7.00 × 10^0^	−3.73 × 10^−1^	−1.82 × 10^−1^
GWO	−2.17 × 10^0^	−9.83 × 10^1^	−1.27 × 10^0^	−1.75 × 10^1^	−7.98 × 10^−1^	−8.52 × 10^−1^	−2.81 × 10^1^	−1.60 × 10^−3^	−7.00 × 10^0^	−8.44 × 10^−1^	−1.09 × 10^−1^
PSO	−2.16 × 10^0^	−9.81 × 10^1^	−1.93 × 10^0^	−1.43 × 10^1^	−7.43 × 10^−4^	−9.41 × 10^−5^	−2.81 × 10^1^	−7.14 × 10^−5^	−7.00 × 10^0^	−3.74 × 10^−1^	−5.87 × 10^−1^
ABC	−1.60 × 10^1^	−2.26 × 10^2^	−1.97 × 10^0^	−1.43 × 10^1^	−1.29 × 10^2^	−2.19 × 10^0^	−2.98 × 10^1^	−3.52 × 10^0^	−3.48 × 10^0^	−1.65 × 10^−1^	−1.80 × 10^−1^

## Data Availability

The source codes of the TLOCTO are publicly available at https://ww2.mathworks.cn/matlabcentral/fileexchange/133382-tlocto.

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
