# Peer review of "Teaching–Learning Optimization Algorithm Based on the Cadre–Mass Relationship with Tutor Mechanism for Solving Complex Optimization Problems"

_biomimetics, 2023, doi:10.3390/biomimetics8060462_

Round 1

Reviewer 1 Report

The manuscript must be corrected according to the attached list of my requirements.

 Minor editing of English language required.

Author Response

Dear reviewer,

Thank you for your letter and the professional comments on our paper titled "Teaching-learning optimization algorithm based on the cadre-mass relationship with tutor mechanism for solving complex optimization problems ". Your constructive feedback has been invaluable in improving and refining our paper, and it has provided important guidance for our research. We have carefully examined the feedback you provided and made rigorous revisions to the manuscript, hoping to gain your approval. Your comments are listed below in italics, while our responses are presented in regular font. Changes/additions to the manuscript are highlighted in yellow throughout the text.

Problem 1:The idea is new. The paper is written in relatively good English, but, e.g., on page 1, lines 13, 15 and 46, page 2, lines 58 and 62. However, some issues need to be addressed.

Answer 1: Thank you for pointing out the language errors in my paper. We have made the change. See page 1 and page 2 of the revised paper, in addition, we have carefully reviewed and corrected other language errors in this paper.

Problem 2:The authors did not work with the literature adequately. In references are missing these different generalizations of TLBO (insert them to the references and insert comments on how to change the original TLBO in the “Introduction” section):

Answer 2:Thank you for your suggestion. It is an oversight on our part, but we have now included a detailed review of the improvement literature on TLBO in the introduction section of our paper. The specific changes are on page 2, lines 82 through 103. These papers have provided us with valuable insights and assistance in our research, and we have described them specifically in my paper.

Problem 3:The authors compared TLOCTO with TLBO and six other optimizers, which is inadequate. It is pointless to compare the success of TLOCTO with TLBO when in all the above given papers, TLBO was generalized, and all these algorithms had better results. So, add a new table to the article comparing TLOCTO with the top ten generalizations of TLBO (see the given list of bibliography above).

Answer 3: Thank you for your constructive suggestions. Based on the original paper, we have incorporated five new algorithms and compared them in CEC2020 (20D) (detailed description in section 4.5.3 of the paper, Page 18 lines 463 through 480). According to the experimental results, our proposed TLOCTO algorithm still achieved the best results.

Problem 4:In this research, the authors used TLOCTO to solve some applied problems and compare it with a group of other metaheuristic algorithms. Analyze much more the results and describe the advantages and disadvantages of your proposed method! Mainly:

  1. Insert a subsection, in which the computational complexity of the proposed TLOCTO is analyzed.
  2. How do your account for constraints when solving engineering problems? Insert detailed description in the paper.
  3. To validate the results, authors must provide MATLAB or Python code of the proposed TLOCTO. They must upload their code files to a repository, such as MathWorks, CODE OCEAN, or GitHub, and include the link to this repository in the manuscript.
  4. The authors used the Wilcoxon rank-sum test, which is meaningless, because they always use the same benchmark function for a pair of compared optimizers, and therefore it is a pair of data, and the Wilcoxon signed-rank test must be used.

Answer 4:

a. Complexity analysis is performed in Section 3.4 of this paper (6 pages, lines 235 to 243).

b. A way to consider constraints when solving engineering problems has been inserted in Section 5 of this article (lines 489 through 498 of 20 pages).

c. The code link to the paper is provided in the abstract section of this paper (lines 23 to 25 on page 1).

d. The test in this article has been replaced with the Wilcoxon signed rank test (Lines 349 through 358 on page 11).

Problem 5:The mathematical symbolism in the article is not right. Correct mathematical notation through the whole paper.

Answer 5: Thank you for pointing out the formatting errors in this article. All formula formats in the paper have been carefully revised according to your suggestions.

We appreciate your warm work earnestly and hope that the correction will meet with approval.

Once again, thank you very much for your comments and suggestions, and we look forward to hearing from you.

With best regards, 

All authors.

Reviewer 2 Report

The article has many flaws, of which I will only comment on the most relevant:

The introduction is a mess, the references do not match the text to which they are linked. Just one example among practically the vast majority of references that are not correct: in [19] there is nothing related to engineering problems.

There are errors in the formulae and many typing errors.

For comparative analysis CEC2020 functions are used, the winner of CEC2020 and later editions should be used.

For engineering problems, some of the solutions provided are infeasible, the constraints must be included and the nature of the variables (e.g. Ni integer) must be respected.

Not only the TLBO, but also variants of the TLBO must be included.

A lot of typos

Author Response

Dear reviewer,

Thank you for your letter and the professional comments on our paper titled "Teaching-learning optimization algorithm based on the cadre-mass relationship with tutor mechanism for solving complex optimization problems ". Your constructive feedback has been invaluable in improving and refining our paper, and it has provided important guidance for our research. We have carefully examined the feedback you provided and made rigorous revisions to the manuscript, hoping to gain your approval. Your comments are listed below in italics, while our responses are presented in regular font. Changes/additions to the manuscript are highlighted in yellow throughout the text.

Problem 1:The introduction is a mess; the references do not match the text to which they are linked. Just one example among practically the vast majority of references that are not correct: in [19] there is nothing related to engineering problems.

Answer 1:Thank you for pointing out the reference errors in my paper, all formula format errors in this article have been corrected, and the textual content of the entire document has also been carefully revised.

Problem 2:There are errors in the formulae and many typing errors.

Answer 2: Thank you for pointing out the formatting errors in this article. All formula formats in the paper have been carefully revised according to your suggestions.

Problem 3:For comparative analysis CEC2020 functions are used, the winner of CEC2020 and later editions should be used. Not only the TLBO, but also variants of the TLBO must be included.

Answer 3: Thank you for your constructive suggestions. Because CEC2020 and later test function suites have not yet announced the winning algorithm of the contest, this paper chooses to compare the TLOCTO algorithm with two other kinds of algorithms on the CEC2020 (Dim=20) test functions: an improved teaching-learning-based optimization algorithm (RLTLBO), and teaching-learning-studying-based optimization (TLSBO), are new variants of the TLBO algorithm, respectively. In addition, the black widow optimization algorithm (BWOA), Archimedes optimization algorithm (AOA), and war strategy optimization algorithm (WSO) are the highest cited algorithms in 2020, 2021, and 2022, respectively (detailed description in section 4.5.3 of the paper, Page 18 lines 463 through 480). According to the experimental results, our proposed TLOCTO algorithm still achieved the best results.

Problem 4:For engineering problems, some of the solutions provided are infeasible, the constraints must be included and the nature of the variables (e.g. Ni integer) must be respected.

Answer 4: Thank you for pointing out the flaws in my engineering experiment. In chapter 5 of this paper, we incorporated the corresponding constraint values for each engineering problem and re-conducted the experiments that did not meet the requirements of the problem (planetary gear train design optimization problem) (Pages 20 to 23).

We appreciate your warm work earnestly and hope that the correction will meet with approval.

Once again, thank you very much for your comments and suggestions, we look forward to hearing from you.

With best regards, 

All authors.

Reviewer 3 Report

The manuscript can be accepted after moderate English polishing.

Need to be polished moderately.

Reviewer 4 Report

This paper brings a novelty to the scientists and researchers in the studied field.  As far as the research content is concerned, it is interesting and suitable for the readers of this journal. As an expert in the field, I consider the methodology presented in the paper interesting, and its effectiveness is proved properly. Therefore, I recommend accepting and publish it in its current form.

Author Response

Dear Reviewer,

Thank you for your valuable feedback on our paper. We are delighted to know that you find our research content interesting and suitable for the readers of this journal. We appreciate your positive remarks regarding the novelty our work brings to the scientific community.We also acknowledge your assessment of the methodology presented in the paper. It is encouraging to learn that you found it interesting and that its effectiveness was adequately demonstrated. Your recommendation to accept and publish our paper in its current form is greatly appreciated.We have carefully considered all your suggestions and comments, and we believe they have significantly improved the quality of our work. We would like to thank you for taking the time to thoroughly review our paper and provide such thoughtful feedback.
Once again, we express our gratitude for your positive evaluation of our paper, and we look forward to seeing it published in this esteemed journal.

Best regards,
Xiao Wu

Round 2

Reviewer 1 Report

I am almost 100% dissatisfied with the correction of the article. The authors did not complete the bibliography requested by me, did not correct the mathematical notation according to my explanation in the previous review, and, most importantly: did not include in the article a new comparison of TLOCBO with at least four other optimizers that are improvements of TLBO (my included list of papers with these optimizers). If the authors do not do this even in the following correction, then I will reject the article because it is completely unpublishable without this more sophisticated comparison.

Reviewer 2 Report

The CEC2020 winners have been published. Note that, the most cited algorithms, unfortunately, do not always coincide with the best ones.

The references are still wrong, and with it the introduction.

No comments

Author Response

Dear Reviewers,
We gratefully thank you for your time and effort that you have put into reviewing the manuscript. Your professional and useful suggestions have enabled us to improve our paper. Each suggested revision and comment brought forward by you was accurately incorporated and considered. Revised portions are marked in yellow in the manuscript. Below your comments are point-to-point responses and the revisions are indicated. We appreciate for your warm work earnestly, and hope that the correction will meet with your approval. Thanks again for your valuable comments, which are very important and instructive for our future research. Looking forward to hearing from you.
Thank you and best regards.
Yours sincerely,
Xiao Wu
E-mail: xiao_wu1999@163.com

Please refer to the attachment for specific modification!
